# Destructive Effect of High-Temperature Heat Flow of Solid Slow-Release Energetic Materials on a Steel Target

**DOI:** 10.3390/ma16041362

**Published:** 2023-02-06

**Authors:** Bingxü Qiao, Zengyou Liang, Fudi Liang

**Affiliations:** School of Mechanical and Electrical Engineering, North University of China, Taiyuan 030051, China

**Keywords:** solid slow-release energetic materials, high-temperature heat flow, destructive effect, theoretical model, numerical simulation

## Abstract

To investigate the high-temperature heat flow’s destructive effect of solid slow-release energetic materials on a steel target, we prepared a sample of solid slow-release energetic materials, eruption devices, and a complete test system to conduct the destruction of high-temperature heat flow on the steel target. In addition, we proposed the energy density to characterise the high-temperature heat flow performance and numerically simulated the destructive effect of the high-temperature heat flow on the steel target. The numerical simulation results were in good agreement with the test results, and the error between them was under 8.5%. Based on the test and simulation results, the steady-state melting model of the steel target was established under the action of high-temperature heat flow. Moreover, a time-varying model of the melting hole shape was found. The results showed that the model of destroying the steel target with the high-temperature heat flow of solid slow-release energetic materials was highly accurate. Therefore, the model could provide theoretical guidance for designing and applying solid slow-release energetic materials in ammunition destruction, metal cutting, the simulation of the laser thermal effect, etc.

## 1. Introduction

Solid slow-release energetic materials are a new kind of energetic materials synthesised by the composite treatment of fuel or oxidants [1]. Because of the low ignition temperature [2,3], fast heat release rate, and high energy release efficiency [4], since their invention, they have attracted the wide attention of researchers at home and abroad. They have been successfully applied in combustion agents, solid propellants, and pyrochemistry. Jia S.Z. et al. summarised the preparation methods of solid slow-release energetic materials and pointed out the advantages and disadvantages of different methods. They analysed the research and application status of solid slow-release energetic materials in military, metal welding, composite manufacturing, and thermal cutting, as well as the research focus and development direction of application and preparation [5]. Sundaram Dilip et al. provided a comprehensive review of the advances in synthesis, properties, and applications of metal-based energetic materials. Low-temperature oxidation processes and the ignition and combustion of metal nanoparticles were investigated. They also studied the burning behaviours of different energetic material formulations with metal particles [6]. Plantier K.B. et al. found that combustion wave speed was a vital function of the stoichiometry of the mixture and a slightly fuel-rich mixture provides an optimum combustion wave speed regardless of the oxidiser synthesis technique [7]. De Souza et al. conducted modelling and simulation of common hematite–aluminium thermite reactions to predict temperature levels and radial burning speeds in a thin disk ignited at the centre [8].

When solid slow-release energetic materials are applied in different fields, some requirements exist for the destructive effect between high-temperature heat flow and the steel target. In the area of waste ammunition destruction, it is required that the melting hole should form quickly when high-temperature heat flow acts on the metal of the projectile body. The gas generated by the combustion of the internal charge can be smoothly discharged so that the ammunition charge’s DDT (deflagration to detonation transition) can be avoided. When disposing of spent munitions, the standard methods used are combustion destruction, explosive destruction, and the decomposition of munitions [9,10]. Yi J.k. et al. summarised the research history and application fields of solid slow-release energetic materials as well as the principles, methods, advantages, and disadvantages of ammunition destruction [11]. They obtained the optimal allocation ratio and reaction compactness from formulation studies of solid slow-release energetic materials [12]. Then, the optimised formula was used to carry out the ablative tests of the steel plates and the combustion destruction tests of simulated ammunition [13]. In metal cutting, high-temperature heat flow was required to quickly melt metal to achieve a quick rescue. Wang P. et al. summarised the research history, current situation, and development trend of pyrotechnic cutting technology at home and abroad. They believed that pyrotechnic cutting technology has many applications in emergency rescue, ammunition destruction, and other aspects [14]. Wang S. et al. proposed and studied a kind of combustion-cutting projectile with solid slow-release energetic material as an energy source. They determined the formula’s composition and analysed the influence of different mix systems on the cutting performance [15]. Wu Y.S. et al. changed the content of the main ingredients of the formula and prepared the combustion electrode to cut Q235 steel plates. As a result, they found the optimal content ratio [16]. When simulating the irradiation effect of laser weapons with different power, producing the high-temperature heat flow generated by solid slow-release energetic materials with varying energy densities is necessary [17].

The focus of the research on the combustion energy release behaviour of solid slow-release energetic materials lies in the study of its formula and manufacturing process, so the destructive effect between the high-temperature heat flow and the target is rarely studied [18,19,20,21]. To realise the safe, reliable, and rapid heating and melting of the target plate with high-temperature heat flow, the destructive effect of the high-temperature heat flow of solid slow-release energetic materials on steel targets must be studied. An accurate model of steel target destruction caused by high-temperature heat flow was established by combining theory, testing, and simulations. It could provide a theoretical basis for accurately controlling solid slow-release energetic materials’ combustion and energy release process. This finding could play a guiding role in solid slow-release energetic materials applications, such as ammunition destruction, metal cutting, and the simulation of the laser thermal effect.

## 2. Experimental Study

The eruption device made of 45# steel is shown in Figure 1. The layout of the test equipment is shown in Figure 2:

We carried out ten tests in a dedicated test site, and the main instruments were the high-speed camera, thermocouple, manometer and infrared cameras, etc. Based on the previous research basi, and the relevant theories and literature ([19,20]), we designed the formula for the test sample. The contents of Ba(NO_3_)_2_ in tests 1–5 were 22%, 24%, 26%, 28%, and 30%, respectively, and the throat size of the eruption device was 50 mm, aiming at controlling the high-temperature heat flow temperature, velocity, and combustion chamber pressure. The throat diameters in tests 6–10 were 40, 45, 50, 55, and 60 mm, respectively, and the content of Ba(NO_3_)_2_ was 26%, aiming at controlling the effective diameter of the high-temperature heat flow.

## 3. Results and Analysis

### 3.1. Combustion of Solid Slow-Release Energetic Materials

In Test 3, the beginning time (0 s) of combustion was set when the sample had just ignited and emitted a bright white light. As seen in Figure 3, the flame ejected from the nozzle after the ignition and acted vertically on the steel target surface. In 2–3 s, smoke was produced, accompanied by the scattering and expansion of reaction products. The combustion of the sample was the most intense between 6 and 7 s. At about 8 s, the steel target was perforated by the high-temperature heat flow, accompanied by a harsh cracking sound (Test Plates 1, 2, 4, and 5 were not perforated). At around 10 s, the intensity of the flame significantly decreased. The combustion duration was about 13 s, and the action time of the high-temperature heat flow on the steel target was about 12 s.

The high-speed and infrared camera images at the moment of the highest heat flow temperature of Tests 2 and 5 are shown in Figure 4. The change curves of the maximum heat flow temperature, the maximum pressure of the combustion chamber, and the maximum velocity of Tests 1–5 are shown in Figure 5, Figure 6 and Figure 7.

The maximum temperature increased from 1845 °C at 22% to 2429 °C at 26%, then dropped to 1987 °C at 30%. These temperature values were similar to those of (about 2000 °C) [22]. The maximum pressure increased from 101,305 Pa at 22% to 108,535 Pa at 26% and then dropped to 99,265 Pa at 30%. The maximum velocity increased from 345.3 m/s at 22% to 430.8 m/s at 26% and then dropped to 190.5 m/s at 30%. Because Ba(NO_3_)_2_ played a dual role as an oxidising and gas-producing agent in the samples, when the content of Ba(NO_3_)_2_ increased from 22% to 26%, both the maximum temperature of the heat flow and the maximum pressure of the combustion chamber increased at different degrees. When the content of Ba(NO_3_)_2_ was excessive, part of Ba(NO_3_)_2_ completely reacted with Al powder, and the remaining Ba(NO_3_)_2_ decomposed under high temperature and the pressure to produce more oxygen and nitrogen. In addition, Al powder underwent a combustion reaction with the oxygen released by Ba(NO_3_)_2_ and the absorbed heat. As the amount of Ba(NO_3_)_2_ and Al powder was fixed in the formula; the higher the content of Ba(NO_3_)_2_, the less the proportion of Al powder and the less apparent the endothermic effect. Therefore, when the content of Ba(NO_3_)_2_ was under 26%, the combustor pressure and the high-temperature heat flow velocity increased with the rise of the content of Ba(NO_3_)_2_. With the increase of Ba(NO_3_)_2_ content from 26% to 30%, the maximum temperature, maximum pressure, and maximum velocity decreased to varying degrees. This phenomenon indicated that the mass proportion of Al powder decreased, leading to an insufficiency of the primary energy source of the reaction. There was an optimal allocation ratio of each test in the sample (Al:Ba(NO_3_)_2_ = 26:24). Increasing or decreasing the content of any component reduced the heat flow energy. The authors also believed that adjusting the oxide and metal powder content could control the heat flow temperature [22].

### 3.2. High-Temperature Heat Flow Energy Density

To uniformly characterise the performance parameters of the high-temperature heat flow generated by the samples, such as the heat flow velocity, heat flow temperature, and pressure of the combustion chamber, the concept of high-temperature heat flow energy density and calculation Formula (1) were proposed for the first time, representing the internal energy contained in the high-temperature heat flow acting on the unit area within a unit of time.
(1)$$q_{in}=\frac{\sum_{i}P_{i}\times A_{i}+P_{g}}{S}$$
where $q_{in}$: energy density; S: action area; $P_{i}$: the power of liquid or solid products; $A_{i}$: the mass fraction of the liquid or solid product; and $P_{g}$: the power of the gaseous product.

We used energy density to characterise the properties of heat flow. The quantitative calculation of this concept required using three primary parameters: combustor pressure, heat flow temperature, and heat flow propagation velocity, so we considered three points. The maximum heat flow velocity, maximum temperature, maximum combustor pressure, and energy density of 10 tests are shown in Table 1:

The state of the Q235 steel targets before and after the tests is shown in Figure 8.

The thickness of each Q235 steel target was 10 mm, and the diameter was 30 mm. WRN thermocouples were arranged at 0, 30, and 50 mm from the centre of the steel target. Figure 8 shows the state of the steel targets before and after the tests. Damage to the steel targets in Tests 1–5 corresponded to the magnitude of the energy density. The sample of Test 3 had a higher energy density (9125 W/cm^2^), corresponding to the most severe damage to the steel targets. The steel targets of Tests 2 and 4 had different degrees of depression. The steel targets of Tests 1 and 5 only showed a slag hanging phenomenon. Perforations occurred on the steel targets of Tests 6–10. See Table 2 for the perforations on the front and back of the steel targets. See Figure 9 for the temperature rise curve of the No. 2 temperature measuring point on the back of the steel targets.

In Tests 6–9, the effective heating diameters and the perforations also correspondingly increased with the increase of the throat diameter of the eruption devices. The diameters of the perforations were similar to those of the throat. In Tests 6–10, the diameter of the throat positively correlated with the heat flow’s action range. With the increased action range of heat flow, the effective action area on the target plate also expanded, so the diameter of the weld hole increased. From the images recorded by the high-speed camera, it could be calculated that the time of heat flow melting through the steel targets was about 4.5–6.2 s, so the average penetration speed was about 1.61–2.22 mm/s. However, the perforation’s diameter on the front of the Test 10 steel target was 58 mm, much lower than that of the eruption device’s throat (66 mm). One of the reasons for this was that on the premise of keeping the inlet diameter, the outlet diameter and total length of the nozzle did not change, and the larger the throat diameter, the smaller the convergence ratio and the size of the convergence section. The more the expansion length increased, the greater the loss of high-temperature heat flow energy and the more the energy density decreased. Therefore, as shown in Figure 9, the heating rate on the back of the steel target was 175.2 °C from 5–10 s, which was lower than Test 8′s 224.3 °C. The energy loss also led to the considerable aperture difference between the front and the back of Test 10′s steel target. In other words, when the diameter of the eruption device throat was greater than 55 mm, there was a significant loss of heat flow energy.

## 4. Numerical Simulation of the Destructive Effect of Heat Flow on the Steel Target

In fluent, we set the metal jet flow and high-temperature gas phase as continuous and the unburned solid particles as discrete. Based on the VOF and DPM models, the numerical simulation of temperature rise, melting, and deformation of solid materials under high-temperature heat flow was realised by stratifying the target plates. The simulation’s initial and boundary conditions were based on the data obtained from the tests, as shown in Table 3.

### 4.1. The Heating Process of Steel Target under the Action of High-Temperature Heat Flow

A numerical simulation was carried out based on the data from Test 8. The temperature distribution of the heat flow is shown in Figure 10. The heat flow from the nozzle acted vertically on the centre of the target plate. The heating process of the steel target is shown in Figure 11. Over time, the hole deepened and eventually perforated at about 5 s. Three temperature measuring points with the same distribution positions as those of the tests were selected on the back of the steel target to extract the temperature curve. The simulation and test results of the temperature rise curve are shown in Figure 12. According to the simulation results, the maximum temperature of Nos. 1, 2, and 3 on the back of the steel target were 1658 °C, 1211 °C, and 558 °C, respectively, and the average heating rate between 2 and 7 s was 303.8 °C⁄s, 157 °C⁄s, and 75 °C⁄s, respectively. The error of the test and simulation results was no greater than 6.5%, so the simulation results had high accuracy and could be used as the basis of theoretical calculation.

### 4.2. The Damage Process of the Steel Target under the Action of the High-Temperature Heat Flow

The velocity distribution of heat flow acting on the steel target is shown in Figure 13. The velocity distribution at the moment of perforation is shown in Figure 14. The shape of the hole was similar to that of the cone. The process of heat flow penetrating the steel target is shown in Figure 15. Over time, the hole deepened and eventually perforated at about 5.5 s.

Compared with the test results, the front perforation was 54.6 mm, which increased by 8.5%; the back perforation was 50 mm, which increased by 4%. The average penetration velocity was 2.17 mm/s, which increased by 2%. The simulation results were generally in good agreement with the test results. The reason for the error was that the velocity and temperature of high-temperature heat flow were inputted as the maximum value, and the value was constant. Therefore, the numerical simulation results were slightly larger than the test results, and the error was less than 8.5%.

## 5. The Destructive Effect Model of High-Temperature Heat Flow on the Steel Target

### 5.1. Steady-State Melting Model

Assuming that the density of the steel target did not change, the solid-liquid interface condition of the model could be obtained as follows:(2)$$\left\{\begin{array}{c} -k_{q}\frac{\partial T_{q}}{\partial z}+k_{s}\frac{\partial T_{s}}{\partial z}=\rho_{t}L_{m}\frac{dZ_{m}}{dt}\\ T_{s}=T_{q}=T_{m} \end{array}\right.t>t_{m},z=Z_{m}\left(t\right)$$
where s and q: solid and liquid states, respectively; m: melting state; *L_m_*: the latent heat of melting; *t_m_*: the time at which melting started.
(3)$$t_{m}=\frac{\pi k_{s}^{2}T_{m}^{2}}{4\kappa_{s}A^{2}q_{in}^{2}}$$

The moving velocity *U_m_* and the position *Z_m_* of the solid–liquid interface were obtained as follows:(4)$${\left(a\right)\;U}_{m}=\frac{Aq_{in}}{\rho \left(C_{s}T_{m}+L_{m}\right)}\;\;\;{\left(b\right)\;Z}_{m}=U_{m}\left(t-t_{m}\right)$$

Combined with the temperature field of the steel target without phase transition under the action of high-temperature heat flow, the temperature field $T_{S}\left(Z\right)$ in the solid state of the steel target after phase transition could be calculated as
(5)$$T_{s}\left(z\right)=T_{m}exp\left[\frac{-Aq_{in}\left(z-Z_{m}\right)}{\rho k_{s}\left(C_{s}T_{m}+L_{m}\right)}\right]\;z\geq Z_{m}$$
where *U_m_*: melting rate; *A*: absorption rate; $q_{in}$: energy density; ρ: the material density of the steel target; *C_s_*: the specific heat capacity of the steel target; *T_m_*: the melting temperature of the steel target; *L_m_*: the latent heat of melting; *k_s_*: thermal conductivity.

### 5.2. Time-Varying Model of Melting Hole Shape

To obtain the melting hole shape of the steel target at a particular moment was equal to getting the penetration depth of the point whose energy density is higher than the critical energy density at that moment (Figure 16). Then, it was also necessary to obtain the distribution of energy density on the surface of the steel target under the condition that the penetration depth formula is known.

Based on the energy density distribution of Tests 7–9, the centre of the steel target was taken as the coordinate origin, and the energy density of the heat flow was extracted every 5 mm. The distribution of the high-temperature heat flow within its effective diameter is shown in Figure 16. The energy density within the diameter of 50 mm was high and evenly distributed. With the increase in distance, the energy density slightly decreased. When the distance from the centre of the steel target was greater than 25 mm, the energy density sharply dropped. The diameter of the melting hole on the front of the steel target was approximately equal to the throat’s diameter. Matlab was used to fit the energy distribution curve. We read the test data and saved them into the Matlab variable. The instruction lsqcurvefit belongs to Matlab’s optimisation toolbox. It uses the least square method to evaluate parameters from the initial guess value and fits the original data points to the nonlinear function. The specific format is x = lsqcurvefit (fun, x_0_, x data, and y data), where x and y data are the original data points; fun (x, x data) is the fitting function; x is the coefficient of the fitting function; x_0_ is the coefficient’s initial guess; and its initial setting may affect the result. If the fitting result is not ideal, the initial value x_0_ can be changed and then fitted again. Moreover, the relationship between the heat flow and the radial distance of the steel target was obtained as follows:(6)$$q\left(r\right)=\left\{ \begin{array}{l} -0.0144\;q_{\max}r^{2}-0.0182\;q_{\max}r+q_{\max}\begin{array}{cc} & 0\leq r\leq r_{p} \end{array}\\ 0.0078\;q_{\max}r^{3}-0.0735\;q_{\max}r^{2}+0.0702\;q_{\max}\begin{array}{cc} r+q_{\max} & r_{p}<r\leq r_{s} \end{array} \end{array}\right.$$
where $r_{p}$: the radius of the throat and $r_{s}$: the radius the of the steel target. Then, combining Equations (4) and (6), the time-varying melting hole shape can be obtained.

### 5.3. Theoretical Model Verification

#### 5.3.1. Verification of the Steady-State Melting Model

The damage model of heat flow on the steel target mainly included the relationship between penetration depth with time, that is, the penetration velocity of the steel target by heat flow and the change of the melting hole shape over time.

The heat flow penetration velocity on the steel target’s central axis in the 7th, 8th, and 9th tests was extracted. The test results were compared with the theoretical results, as shown in Table 4.

The error of the theoretical calculation results was under 8% compared with those of the test results, which proved the accuracy of the theoretical calculation. Therefore, the steady-state melting model could accurately describe the damage process of the steel target caused by heat flow.

#### 5.3.2. Verification of the Time-Varying Model of Melting Hole Shape

Based on the energy density distribution obtained in test No. 8, with the centre of the steel target as the coordinate origin, the calculation points were extracted every 2 mm. The penetration depth of these points was obtained at t = 2 s, t = 4 s, t = 6 s, and t = 8 s. Using Matlab, the depth of the melting hole was fitted into the contour maps and compared with the simulation results, as shown in Figure 17.

The blue and red lines are the fitting and testing results, respectively. The time-varying melting hole shape obtained by the theoretical calculation was highly consistent with that of the simulation, which proved that the theoretical model had high accuracy.

## 6. Conclusions

(1)A complete combustion and energy release test system for solid slow-release energetic materials was designed, and the test parameters such as combustion time, combustor pressure, high-temperature heat flow velocity, and temperature were obtained. The high temperature-heat flow performance could be accurately quantified by the energy density and calculation formula proposed in this paper.(2)The high-temperature heat flow had a significant energy loss when the diameter of the eruption device’s throat was larger than 55 mm.(3)The numerical simulation of the heating and deformation process of solid materials under the impact of fluid materials was carried out in fluent, and the model of high-temperature heat flow of gas–liquid–solid coexistence was realised. We simulated the heating and penetration processes of the steel target under the action of high-temperature heat flow. The numerical simulation results were in good agreement with the test results, and the error between them was under 8.5%.(4)Based on the steady-state melting model, we calculated the penetration velocity of high-temperature heat flow on the central axis of the steel target. The results error between the test and model calculation was under 8%. The distribution function of the energy density of the high-temperature heat flow on the radial direction of the steel target was fitted with Matlab to obtain the time-varying model of melting hole shape. The comparison between the calculation results of the model and the simulation results showed that the model’s accuracy was high.

## Figures and Tables

**Figure 1 materials-16-01362-f001:**
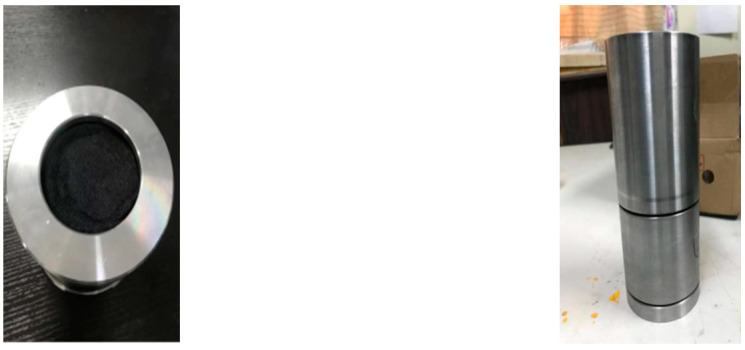
Eruption device.

**Figure 2 materials-16-01362-f002:**
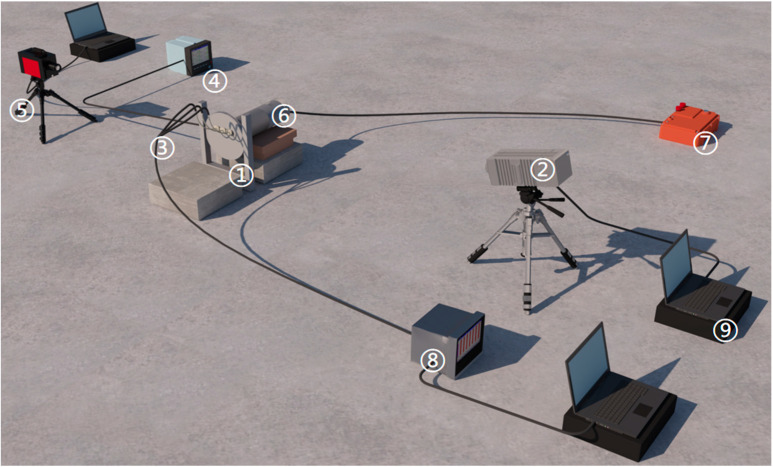
The layout of test equipment: (1) steel target, (2) high-speed camera, (3) thermocouple and thermal compensation wire, (4) internal electronic piezo gauge and recorder, (5) infrared camera, (6) eruption device, (7) trigger, (8) temperature recorder, (9) computer.

**Figure 3 materials-16-01362-f003:**
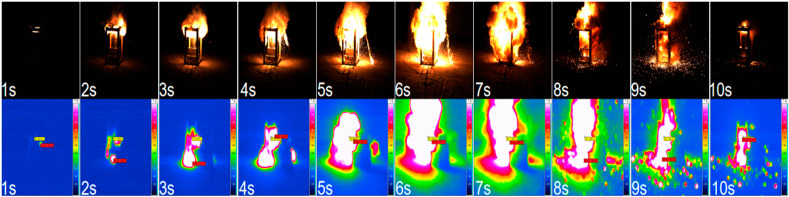
The high-speed camera and infrared camera images of the heat flow growth process of Test 3.

**Figure 4 materials-16-01362-f004:**
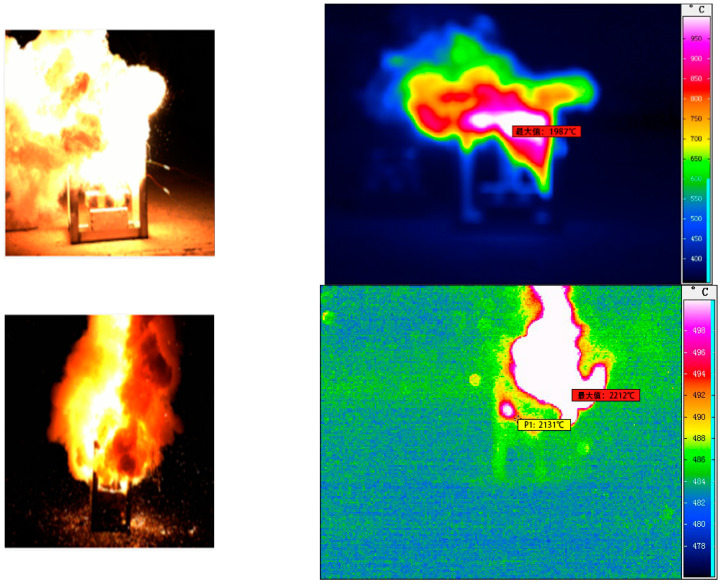
Images of the high-speed camera and infrared camera of the highest temperature moments of Tests 2 and 5.

**Figure 5 materials-16-01362-f005:**
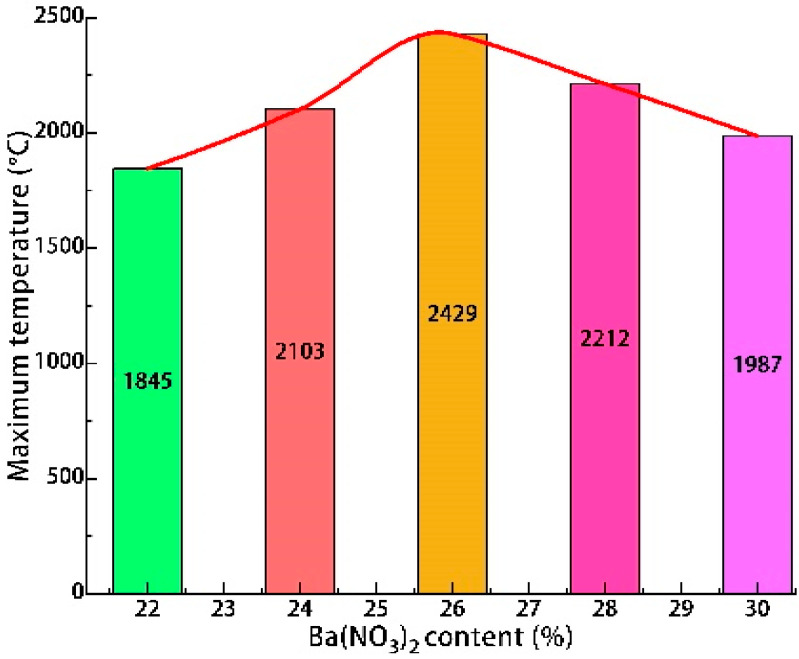
Effect of Ba(NO_3_)_2_ content on maximum temperature.

**Figure 6 materials-16-01362-f006:**
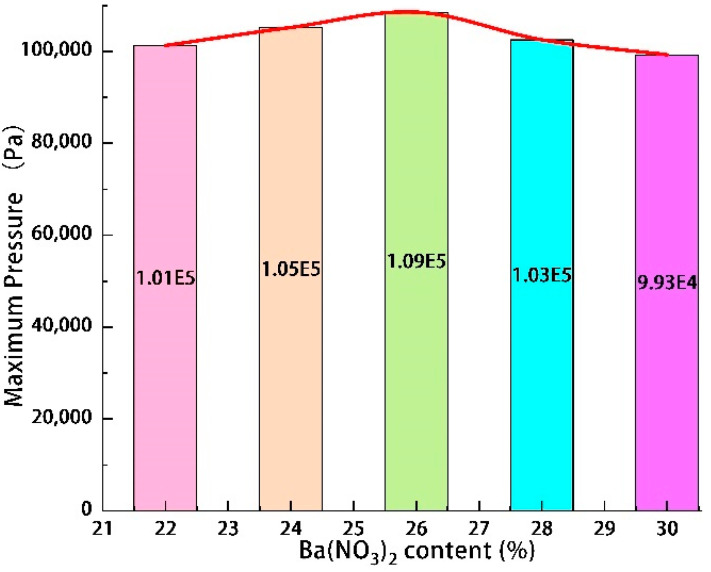
Effect of Ba(NO_3_)_2_ content on maximum pressure.

**Figure 7 materials-16-01362-f007:**
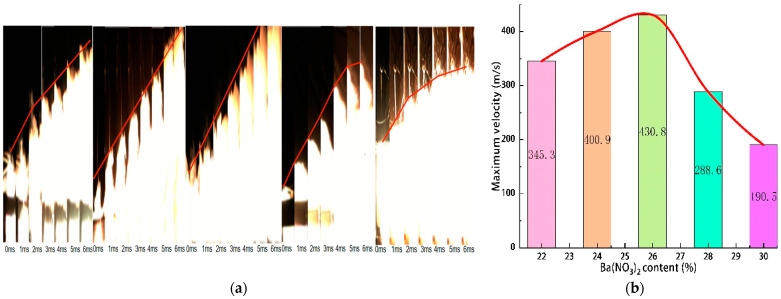
Displacement and velocity changes of heat flow at the axis of the eruption device with different Ba(NO_3_)_2_ content. (**a**) Displacement of heat flow, (**b**) Velocity changes of heat flow.

**Figure 8 materials-16-01362-f008:**
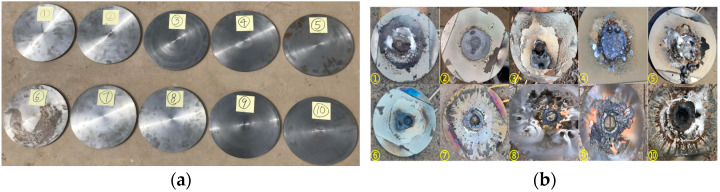
The state of the steel targets before and after the tests. (**a**) Before tests, (**b**) After tests.

**Figure 9 materials-16-01362-f009:**
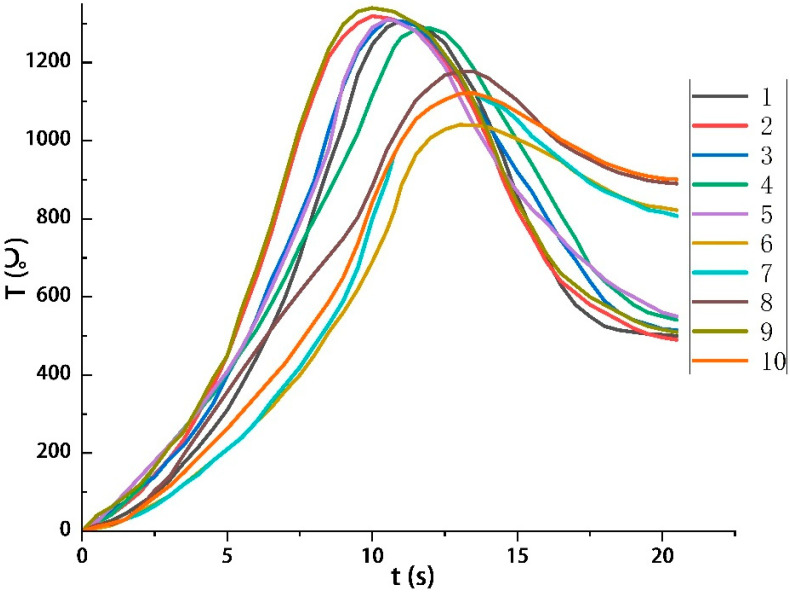
Temperature rise curve on the back of the steel targets of the No. 2 measuring point.

**Figure 10 materials-16-01362-f010:**
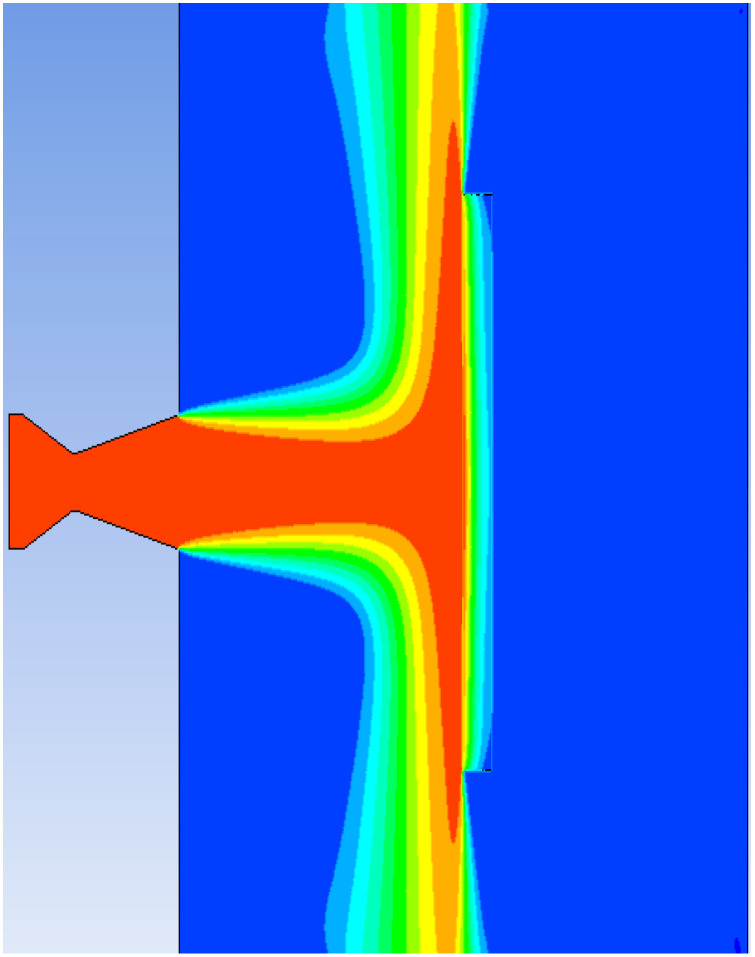
The temperature distribution of heat flow.

**Figure 11 materials-16-01362-f011:**
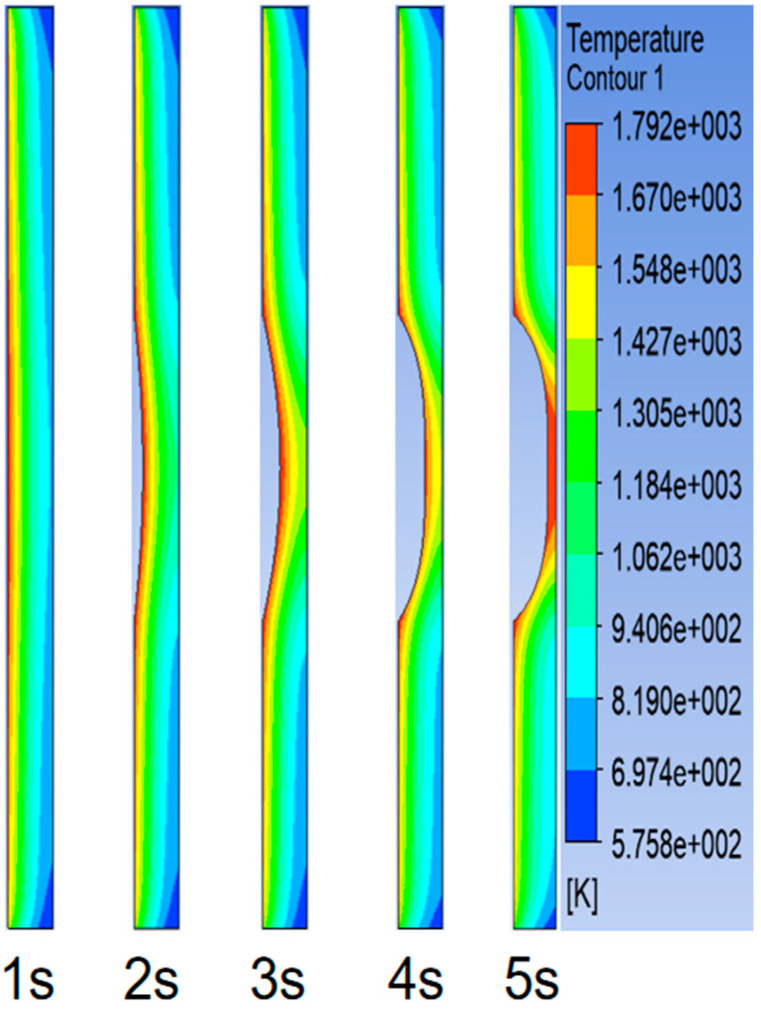
The heating process of the steel target.

**Figure 12 materials-16-01362-f012:**
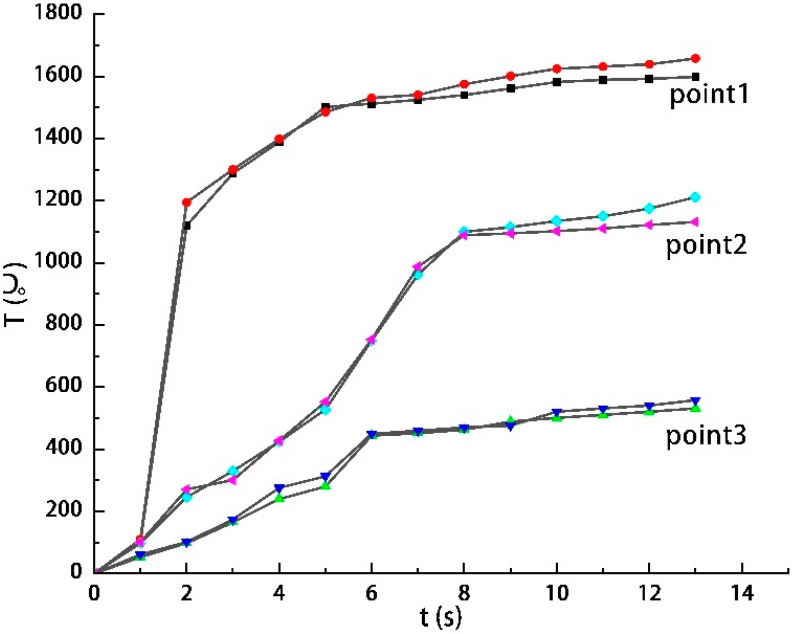
Simulation and test results of the temperature rise curve.

**Figure 13 materials-16-01362-f013:**
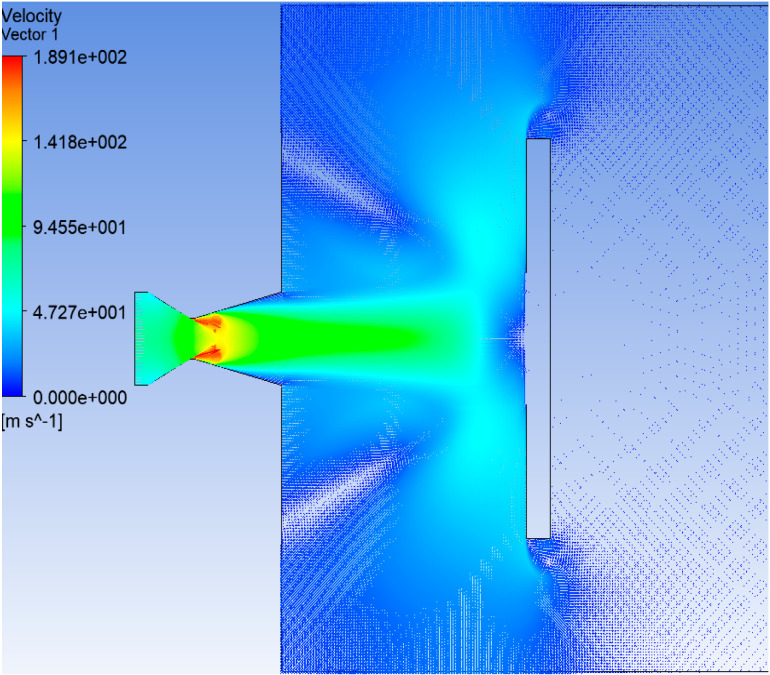
Distribution of heat flow velocity.

**Figure 14 materials-16-01362-f014:**
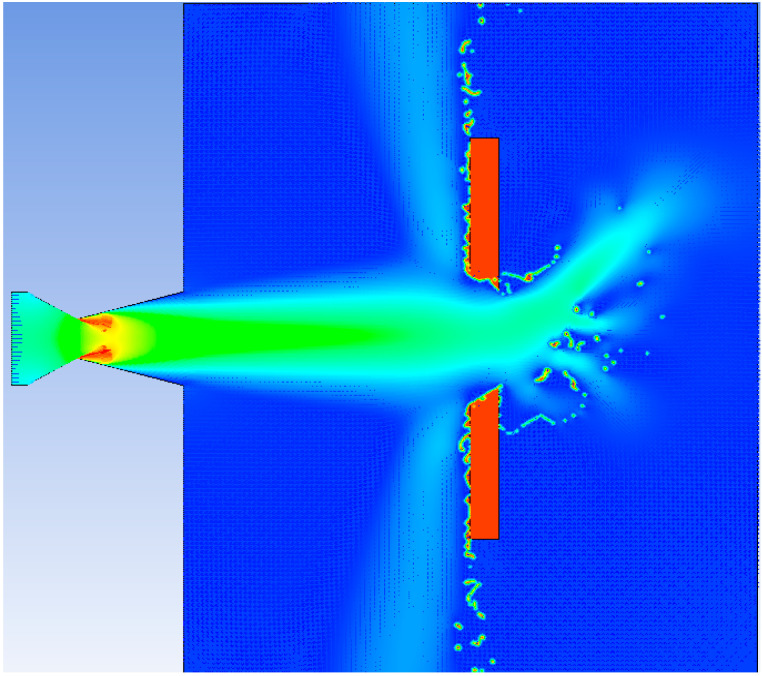
The velocity distribution at the moment of perforation.

**Figure 15 materials-16-01362-f015:**
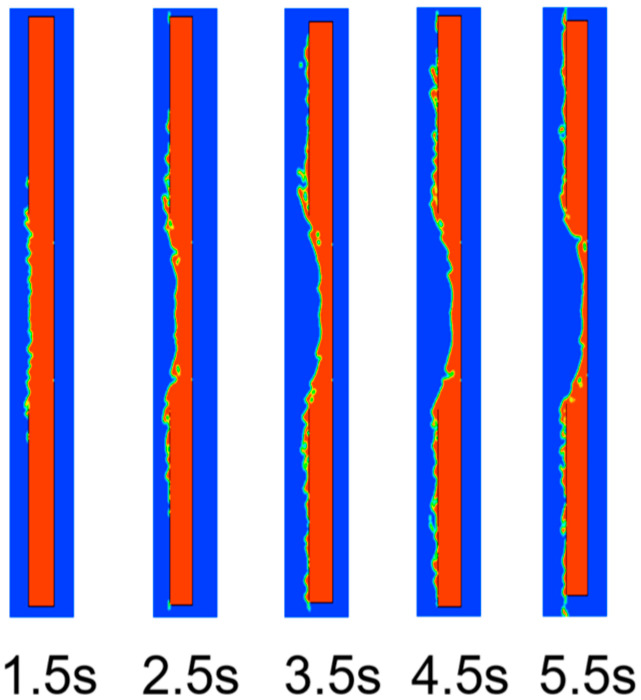
Process of heat flow penetrating the steel target.

**Figure 16 materials-16-01362-f016:**
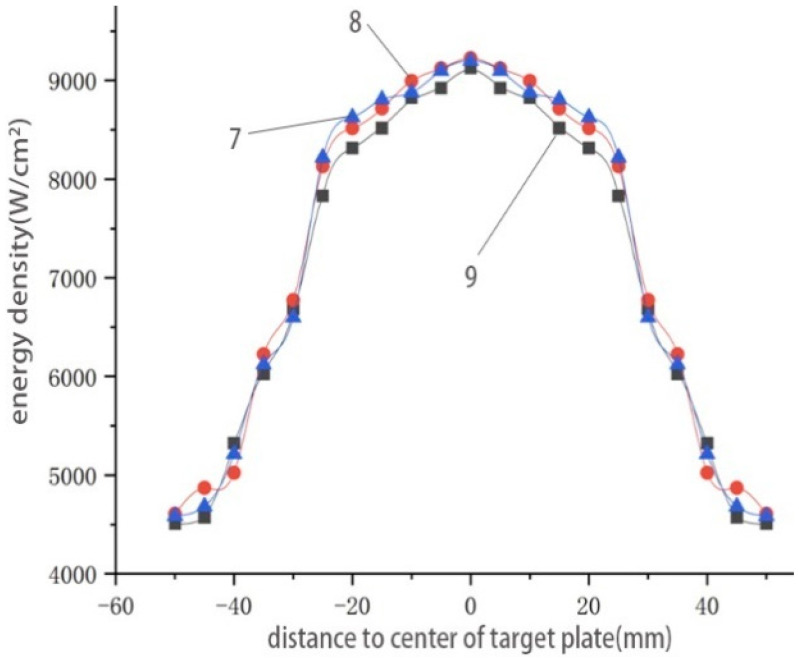
Energy density distribution on the steel target.

**Figure 17 materials-16-01362-f017:**
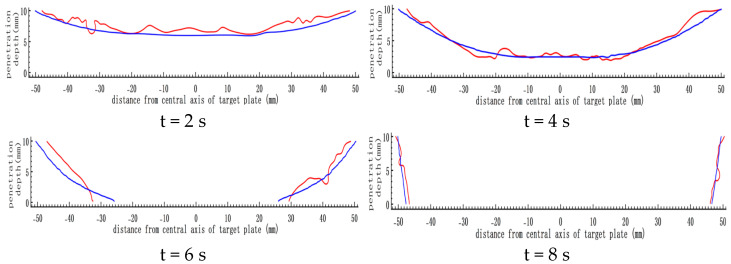
Comparison of theoretical calculation and simulation results. (The blue and red lines are the fitting and testing results, respectively.)

**Table 1 materials-16-01362-t001:** Performance parameters of high-temperature heat flow.

Sequence Number	Maximum Heat Flow Velocity/(m/s)	High-Temperature Maximum Heat Flow Temperature/°C	Maximum Combustor Pressure/Pa	Energy Density/(W/cm^2^)
1	345.3	1845	101,305	7321
2	400.9	2103	105,218	8266
3	430.8	2429	108,535	9125
4	288.6	2212	103,548	6611
5	190.5	1987	99,265	4031
6	424.4	2117	106,162	9591
7	433.9	2283	106,813	9413
8	454.7	2359	107,574	9817
9	405.6	2006	104,976	8997
10	400.1	2136	104,543	8704

**Table 2 materials-16-01362-t002:** Perforation sizes of the steel targets.

Serial Number	Front Aperture	Back Aperture
6	39 mm	35 mm
7	46 mm	43 mm
8	50 mm	48 mm
9	56 mm	55 mm
10	58 mm	49 mm

**Table 3 materials-16-01362-t003:** Initial conditions for numerical simulation.

The Average Density of Heat Flow	Thermal Conductivity of Heat Flow	Specific Heat Capacity of Heat Flow	Turbulence Model	Turbulence Intensity
5.17 g/cm^3^	283.5 W/cm/°C	0.52 J/(g·K)	K-ε	5%
**Thermal conductivity of steel target (solid phase)**	**Thermal conductivity of steel target (liquid phase)**	**Specific heat capacity of steel target (solid phase)**	**Specific heat capacity of steel target (liquid phase)**	**Mass flow rate of solid particles**
260 W/cm/°C	400 W/cm/°C	0.6 J/(g·K)	0.67 J/(g·K)	3.5 g/s

**Table 4 materials-16-01362-t004:** Results of comparison.

Serial Number	Melting Time/s	Energy Density/(W/cm^2^)	Penetration Rate/(mm/s)		
			Test Results	Results of Theoretical Calculation	Error/%
7	4.6	9413	2.17	2.25	3.6
8	4.5	9817	2.22	2.28	2.7
9	5.0	8997	1.96	2.13	8.0

## Data Availability

Data supporting the results presented in this paper will be provided by the corresponding author upon reasonable request.

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
