# Peer review of "Destructive Effect of High-Temperature Heat Flow of Solid Slow-Release Energetic Materials on a Steel Target"

_materials, 2023, doi:10.3390/ma16041362_

Round 1

Reviewer 1 Report

The manuscript seems to be interesting, especially to does working in the  field of using explosives to investigate effects on materials. It is mostly well written, however the biggest concern must be focused to methodology. I must be improved with respect to  materials that were used (better characteristics), and simulation methods (there is nothing writing on the simulation method, on preparation of a simulation test, and it is quite important part of the experiment and discussion). Besides, the results of simulations should be more discussed in terms of obtained experimental results.

Basically, there is a lot of information that should be more discussed in text. The authors should recall experimental results and simulation more often to present the whole view.

Some detail information regarding the reviewed text of manuscript:

1.

line27 -  'pyrochemical something (noun)' or perhaps just 'pyrochemistry'

lines 113 to 115 - the word less is used to often;

line152-153 - grammar should be corrected

2. Experimental method:

It would be valuable to provide information on: what material was used for eruption device, what are the dimensions of this. Besides, what is the thickness of steel plate, what type of steel is used?
Similarly the basics of operation of test site should be explained in text.
It would be interesting to show were thermocouples are placed, perhaps a schematic diagram or a real picture.
How many and what type of thermocouples were used?

In 4.1 the authors mention simulation. It should be explained how the simulation was made? What software? This should be also in experimental method

Also in methodology, the authors should mention Matlab, and explain what was it used for, if applicable, what Matlab tool was used?

3. Comments to Figures:

The images are insufficiently described in text of manuscript.

The authors should consider mentioning required Figures in text. Especially in places where is larger discussion presented.

Generally, descriptions in images are not clear (Fig 7,8,9)

Many Figures should be divided into a), b)... And respectful descriptions should be added. Examples:  Fig 7, 8, 16, 17.

In Fig.8 The numbers of samples after tests are too small.

In Fig. 16 What is presented on the first (a) image?

4. In conclusion:

The authors mention Fluent - It was never mentioned in the manuscript, up to this point.

5. At the end note that authors use various standards for citing authors of articles e.g.:
Jia S.Z. et al.
Sundaram Dilip et al.
De Souza et al.
Yi J.k. et al.

The journal names in the references should be standardized

Author Response

Dear reviewer, thank you for using your precious time to review our paper. Your advice is valuable and useful. According to the suggestion, we have made a careful revision. As follows:

1. Introduction

Line 28 - changed "pyrochemical" to "pyrochemistry"

Line 118 - changed "less than" to "under". line 188 - changed "less than" to "no great than"

Lines 164-167 - changed "Therefore, from 5s to 10s, the heating rate on the back of the steel target was 175.2℃/s, which was much lower than 224.3℃/s of test 8. The energy loss also led to the considerable aperture difference between the front and back of the steel target of test 10. " to "Therefore, the heating rate on the back of the steel target was 175.2℃/s from 5s to 10s, which was lower than test 8's 224.3℃/s. The energy loss also led to the considerable aperture difference between the front and the back of test 10's steel target. "

2. Experimental method

Line 80 - the material of the eruption device was given.

Lines 82 to 85 - the throat size of the erupting device was given.

Line 141 - the material and diameter of the steel target were added.

Lines 141 to 143 - the type, number and location of thermocouples were supplemented.

Lines 171 to 176 - added the software and methods used in numerical simulation.

Lines 230 to 237 - added the methods used in Matlab.

3.

Figure 7 - added subscripts (a) and (b).

Figure 8 - â‘ the number of target plates in Figure 8 was supplemented. â‘¡added subscripts (a) and (b).

Figure 16 - deleted photo (a) in Figure 16.

Line 102 - added a reference to Figure 7.

Lines 105 to 108 - the information in Figure 5, Figure 6 and Figure 7 were further described.

Line 143 - added a reference to Figure 8.

Line 149 - added a reference to Figure 9.

Lines 164 to 167 - the information in Figure 9 was further described.

Line 225 - added a reference to Figure 16.

Lines 225 to 228 - added more description to Figure 16.

Line 258 - added a reference to Figure 17.

Line 259 - added more description to Figure 17.

4.

Lines 171 to 176 - added the software and methods used in numerical simulation.

5.

Fixed formatting problems when referring to author names.

Standardised journal names in the references.

Thank you again for your review, and we welcome your more valuable suggestions for revision.

Reviewer 2 Report

The article describe the destructive effect of high energy heat flow on the steel target. The topic is interesting and has industrial applications, however in my opinion it should be published in the journal related with heat transfer, not in Materials. Also, the background of the study is presented in a chaotic and difficult to understand way. Probably also due to necessary English grammar corrections. Many references are printed in Chinese and are difficult to verify, and it should be changed for available one.

More detailed suggestions and remarks are presented below. 

I suggest changing the title of the article to: Destructive Effect of High Temperature Heat Flow of Slow-release Energetic Materials on a Steel Target.

Abstract: “To investigate the destructive effect of high-temperature heat flow of solid slow-release energetic materials on the steel target, prepared the sample of solid slow-release energetic materials,  eruption devices and complete test system to conduct the destruction of high-temperature heat flow on the steel target.” This and two more sentences in the abstract need revision. The sentences are too long and difficult to understand.

Lines 23-41: Introduction. The introduction is too general, what is the point, goals and purpose. It should be clarified.

Line 48: DDT – needs explanation.

Experimental study: There is no information about chemical composition of materials used for the tests and no description about numerical simulations (boundary conditions, assumptions, and more).

Line 83: The influence of two main parameters has been studied, but there are more. There is no explanation and discussion why the only these two were chosen?

Line 97: Why the test plates 1, 2, 4, and 5 were not perforated? There should be an explanation.

Table 1. If I am right the test number of 3 and 8 have the same throat diameter of 50 mm and 26% of the Ba(NO­3)2 content, therefore the results of both tests should be similar. But in the table 1 there are not. Can you explain it why?

Line 144: What is the steel target made of? What kind of steel?

Fig. 8. Why do the steel targets before testing look dirty (rusty)? How was the surface of the steel target prepared?

Line 147: There is description about the target, but there is no explanation about phenomena occurred. There should be discussion about differences observed of the No. 1-5 and 6-10.

Lines 151: Can you also explain how the perforation was measured?

Lines 174-183: The text is difficult to understand. The numerical simulation was carried out based on the data from test No. 8, whereas in Figure 12 there are curves of the maximum temperature of No. 1, No. 2 and No. 3?

Line 195: “Therefore, the numerical simulation results were slightly larger than the test results”. How slightly? This is not precise description.

Author Response

Dear reviewer, thank you for using your precious time to review our paper. Your advice is valuable and useful. According to the suggestion, we have made a careful revision. As follows:

1.Title:

Lines 2 to 3 - changed the title to Destructive Effect of High-Temperature Heat Flow of Slow-release Energetic Materials on a Steel Target.

2.Abstract:

Lines 8 to 9 - changed "To investigate the destructive effect of high-temperature heat flow of solid slow-release energetic materials on the steel target" to "To investigate the high-temperature heat flow's destructive effect of solid slow-release energetic materials on the steel target"

Lines 13 to 14 - added "The numerical simulation results were in good agreement with the test results, and the error between them was under 8.5%."

3.Introduction:

Line 49 - added the explanation of DDT. DDT(Deflagration to Detonation Transition)

Lines 69 to 73 - stated my views on previous studies and the purpose of this paper.

4.Experimental study:

Line 80 - added the material of the eruption device

Lines 82 to 86 - added the throat size of the erupting device

Line 141 - added the material and diameter of the steel target 

Lines 141 to 143- the type, number and location of thermocouples were supplemented

Lines 171 to 176- added the software and methods used in numerical simulation 

Lines 230 to 237- added the methods used in Matlab

Table 3 was added to describe the initial conditions of the numerical simulation.

Dear reviewer, your opinion is correct. We used the concept of energy density to characterise the properties of heat flow. The quantitative calculation of this concept required using three primary parameters: combustor pressure, heat flow temperature and heat flow propagation velocity, so we only considered three points. We revised the manuscript (Lines 135 to 137).

Lines 112 to 125 - An explanation of why the target plates of tests 1, 2, 4 and 5 are not perforated: when the content of Ba(NO3)2 was excessive, part of Ba(NO3)2 completely reacted with Al powder. The remaining Ba(NO3)2 decomposed under high temperature and pressure to produce more oxygen and nitrogen. In addition, Al powder would undergo a combustion reaction with the oxygen released by Ba(NO3)2 and absorb heat. As the amount of Ba(NO3)2 and Al powder was fixed in the formula, the higher the content of Ba(NO3)2, the less the proportion of Al powder and the less apparent endothermic effect. Therefore, when the content of Ba(NO3)2 was under 26%, the combustor pressure and the high-temperature heat flow velocity increased with the rise of the content of Ba(NO3)2. With the increase of Ba(NO3)2 content from 26% to 30%, the maximum temperature, maximum pressure and maximum velocity decreased to varying degrees. This phenomenon indicated that the mass proportion of Al powder decreased, leading to an insufficient of the primary energy source of the reaction. There was an optimal allocation ratio of each test in the sample (Al:Ba(NO3)2= 26:24). Increasing or decreasing the content of any component would reduce the heat flow energy. So the heat flow energy of tests 1, 2, 4 and 5 was weak, resulting in the unperforated target plate. 

Table 1 - Dear reviewer, your opinion is correct. The conditions of tests No. 3 and No. 8 were set up the same, and the heat flow performance of the two should be similar. The main reason for the difference was that test No. 8 was conducted in spring when the climate was expected, while Test No. 3 was conducted in winter when the temperature was low. Low temperature impacted the mass and heat transfer of heat flow in Test No. 3. Although small, the instrument measurement error and sample difference could not be ignored. We were also aware of this problem and would take the initiative to correct it in the follow-up tests to avoid similar issues from happening again. 

Line 141 - added the material and diameter of the steel target 

Dear reviewer. Before the test, we applied a small amount of rust inhibitor on the surface of the target plate to prevent it from rusting. When the photo was taken, a small amount of rust inhibitor was left on it. We cleaned it up during the test. Please forgive us for any misunderstanding.

Lines 112 to 125 - provided the reasons for the differences in the results of tests 1~5. Lines 153 to 156 - added the reasons for the differences in the results of tests 6~10 

The SP2302 profiler measured the dimension of the perforation.

Lines 141 to 143 - I'm sorry. We did not express this clearly in the manuscript, which caused you a misunderstanding. The type, number and location of thermocouples were supplemented. So, there are three temperature rise curves.

Lines 200 to 201 - Adopted a detailed description: "Therefore, the numerical simulation results were slightly larger than the test results, and the error was less than 8.5%."

Thank you again for your review, and we welcome your more valuable suggestions for revision.

Reviewer 3 Report

Present manuscript deal with experimental study and numerical simulation of the destructive effect of high-temperature heat flow of solid slow-release energetic materials on the steel target. Scientific novelty is soundness. The quality of the manuscript is high enough for its publication in Materials.

There are some comments:

1. Results with numerical values should be added in the abstract.

2. Introduction contains general information it can be reduced by 2 times.

3. Discussion of experimental study is very poor. There is no information about composition of solid slow-release energetic material, steel target, parameters of high-temperature heat flow, interaction conditions of last one with steel target, errors of experimental methods, repeatability of experimental data.

4. The quality of figs. 3, 4, 7 is not high enough. Please add temperature scales for infrared camera images. Some of the captions on the figures are too small.

5. Figure 9 is not informative (the thickness of the lines is small, 10 different colors merge).

6. Additional explanations in the figs. 10, 11, 13–15 should improve the perception of the results.

7. Mathematical model takes into account processes occurring on the solid-liquid interface. What about heat and mass transfer in condensed phase and gas area?

8. Initial conditions and constants should be added for numerical simulation.

9. Results with numerical values should be added in the conclusions.

Author Response

Dear reviewer, thank you for using your precious time to review our paper. Your advice is valuable and useful. According to the suggestion, we have made a careful revision. As follows:

1.

Lines 13 to 14- the numerical simulation results were added to the abstract.

2.

Dear reviewer, thank you very much for taking time out of your busy schedule to review our paper. I agree with you. The first paragraph of the introduction described the research progress of the solid slow-release energetic materials, the second paragraph described its application progress, and the third paragraph pointed out the deficiencies of current research and application as well as the purpose and innovation of this paper. We are carefully considering cutting out part of the introduction if we must.

3.

â‘ Supplemented the discussion of experimental studies

Lines 105 to 108- the information in Figure 5, Figure 6 and Figure 7 were further described.

Lines 164 to 167- the information in Figure 9 was further described.

Lines 225 to 228- added more description to Figure 16.

Line 259- added more description to Figure 17.

â‘¡

Dear reviewer, I agree with you. I am very sorry. Due to technical protection, only aluminium powder and Ba(NO3)2 are mentioned in the paper. If you want further information about the material, we can provide it to you.

Line 80 - added the material of the eruption device.

Lines 82 to 86 - the throat size of the erupting device was given.

Line 141 - the material and diameter of the steel target were added.

Lines 141 to 143 - the type, number and location of thermocouples were supplemented.

See table 1 for the parameters of high-temperature heat flow.

Line 201 - the error of the test and simulation results was no greater than 8.5%.

Lines 195 to 201- Compared with the test results, the front perforation was 54.6mm, which increased by 8.5%; the back perforation was 50mm, which increased by 4%. The average penetration velocity was 2.17mm/s, which increased by 2%. The simulation results were generally in good agreement with the test results. The reason for the error was that the velocity and temperature of high-temperature heat flow were input as the maximum value, and the value was constant. Therefore, the numerical simulation results were slightly larger than the test results. The test error is generally small, and the test data repeatability is good.

4.

Added the higher quality images and the temperature scales. Then, increased the size of the captions on the figures.

Figure 7- added subscripts (a) and (b).

Figure 8- â‘ the number of target plates in Figure 8 is supplemented. â‘¡added subscripts (a) and (b).

Figure 16- deleted photo (a) in Figure 16.

5.

In order to improve the accessibility of information, a more precise figure 9 was generated, and the scale and line thickness were adjusted.

6.

Lines 179 to 181- further described the information in the figure.

Lines 192 to 194- further described the information in the figure.

7.

Dear reviewer, I agree with you. I am very sorry. As this paper focused on the damage of heat flow to the target plate, the heat and mass transfer in the condensing phase and gas zone were not described much in the numerical simulation part during manuscript preparation. We intend to analyze these issues in detail in another forthcoming paper.

8.

Table 3 was added to describe the initial conditions of the numerical simulation.

9.

Lines 274 to 275- numerical simulation results were added to the conclusion.

Thank you again for your review, and we welcome your more valuable suggestions for revision.

Round 2

Reviewer 1 Report

The manuscript is improved, however I still miss descripton of the experimental site (in words, not just an image).

In Fig 3, it seems that the sets of images is doubled.

I believe, the geometry of images in Fig 4 is affected. It is a good convenience to keep symmetry and scale. In my opion the images in Fig 4 are expanded in hight while width is kept - comparing to images in version 1 of the manuscript.
Additionally, description of axes in Fig 5 and 6 are difficult to read.

In Table 1, there should be Pa, with capital P

line 146 - spelling, should be 'target'

line 149 - I am thinking if the word 'order' is really proper here? perhaps 'magnitude' would be better? for consideration.

line 235 'Read the Excel data file generated by the experimental instrument and saved it to Matlab’s variables.' is difficult to understand, grammar

Author Response

Dear reviewer, thank you for using your precious time to review our paper. Your advice is valuable and useful. According to the suggestion, we have made a careful revision. As follows:

Lines 82-83 - added test sites and main instruments in words.

Figure 3 - Thank you for your suggestion. You are right. The last time we modified the image, we used a more explicit version and enlarged the size of the image, which is the clearest version we can provide considering the space utilization. If necessary, we will make further improvements.

Figure 4 - Sorry, in the last modification, the picture scale was not appropriate. So we adjusted the ratio of height to width.

Figures 5, 6 and 7 - enhanced the clarity of the axes.

Table 1 - the unit Pa was corrected.

Line 146 - the spelling was corrected to "target".

Line 148 - changed "order" to "magnitude".

Line 231- changed "Read the Excel data file generated by the experimental instrument and saved it to Matlab’s variables." to "Read the tests’ data and saved it into the Matlab variable."

Thank you again for your review, and we welcome your more valuable suggestions for revision.

Reviewer 2 Report

The manuscript was improved, indeed. I have no further comments. However, double-check the spelling and typing errors of the final version before publishing.

Author Response

Dear reviewer, thank you for using your precious time to review our paper. Your advice is valuable and useful. According to the suggestion, we have made a careful revision. As follows:

According to your comments, we have checked the full text for spelling and grammar several times. Your opinion is correct and valuable, which benefits our paper publication.

Thank you again for your review, and we welcome your more valuable suggestions for revision.
